# CI-VAE: a Class-Informed Deep Variational Autoencoder for Enhanced Class-Specific Data Interpolation

## ABSTRACT

We proposed Class-Informed Variational Autoencoder (CI-VAE) to enable interpolation between arbitrary pairs of observations of the same class. CI-VAE combines the general VAE architecture with a linear discriminator layer on the latent space to enforce the construction of a latent space such that observations from different classes are linearly separable. In conventional VAEs, class overlapping on the latent space usually occurs. However, in CI-VAE, the enforced linear separability of classes on the latent space allows for robust latent-space linear traversal and data generation between two arbitrary observations of the same class. Class-specific data interpolation has extensive potential applications in science, particularly in biology, such as uncovering the biological trajectory of diseases or cancer. We used the MNIST dataset of handwritten digits as a case study to compare the performance of CI-VAE and VAE in class-specific data augmentation. We showed that CI-VAE significantly improved class-specific linear traversal and data augmentation compared with VAE while maintaining comparable reconstruction error. In a study of Colon cancer genomics data, we showed that the interpolation between normal cells and tumor cells using CI-VAE may enhance our understanding of cancer development.

## 1 INTRODUCTION

Variational Autoencoders (VAEs) have emerged as a popular unsupervised probabilistic neural network models Kingma & Welling (2013); Dilokthanakul et al. (2016); Hsu et al. (2017) with a variety of applications in computer vision Hsieh et al. (2018); Vahdat & Kautz (2020); Tabacof et al. (2016); Huang et al. (2018), natural language processing Wu et al. (2019); Bahuleyan et al. (2017); Semeniuta et al. (2017), genomics and precision medicine Grønbech et al. (2020); Minoura et al. (2021) and many other domains. In VAEs, through the encoder, a probabilistic latent representation of data in lower dimensional space is inferred. Besides many applications of dimensionality reduction using VAEs, these probabilistic models have proven to be very effective in synthetic data generation. Although VAEs are initially designed for unsupervised learning, several variants of VAEs have been designed in other domains such supervised learning and semi-supervised learning Kameoka et al. (2019); Gómez-Bombarelli et al. (2018); Ehsan Abbasnejad et al. (2017); Xu et al. (2017); Wu et al. (2019); Sohn et al. (2015); Higgins et al. (2016); Zhao et al. (2019).

Suppose we would like to know how two observations transform from one to another. Through linear traversal on the latent space in VAEs, we can generate a trajectory and observe how this transformation may take place. A wide variety of applications can be benefited from this type of solution. For instance, one may attempt to uncover the disease mechanism of Parkinson's through understanding how a neuronal cell in a healthy brain tissue transformed to a neural cell with Parkinson's disease Hook et al. (2018); Blauwendraat et al. (2020). Such investigations are often intended for a specific subset/class of the data. In a Parkinson's disease study, for example, we may intend to investigate the neuronal cells as a cell-type/class of the entire population of cells. Therefore, to perform linear traversal within neuronal cells, we need a latent space that is linearly separable among classes to ensure that, during linear traversal, there is no overlapping of classes (cell types in this example).

With this motivation, we proposed Class-Informed Variational AutoEncoders (CI-VAE), a novel deep learning model architecture that includes an additional linear discriminator applied on the latent space, extending the capabilities of VAEs to form a latent space where observations from different

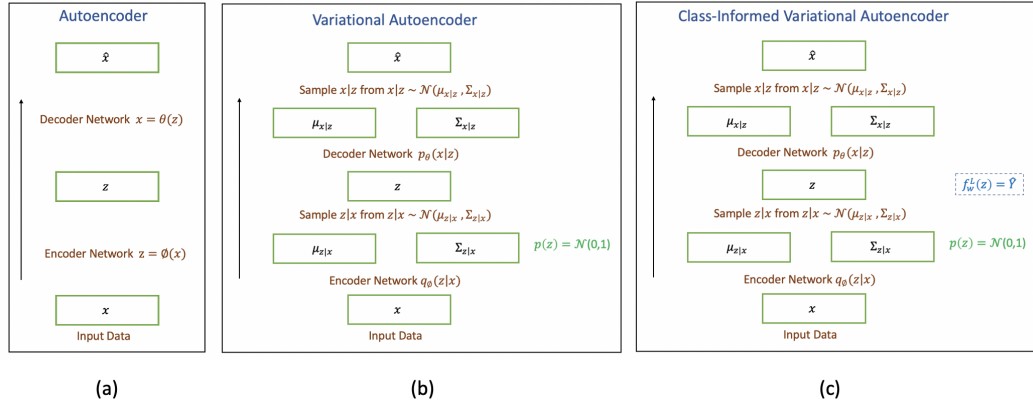

Figure 1: **a.** Autoencoders(AEs): Input data $x$ is deterministically encoded to a lower-dimensional latent variable $z$ and then mapped via decoder to reconstruct the original data $\hat{x}$. **b.** Variational Autoencoder (VAEs): Input data is mapped to the parameters (mean and standard deviation) of the probability function of lower-dimensional latent variable $z$. $z$ is then forming the inferred probability function and the decoder maps the data back to its original form $\hat{x}$. **c.** Class-Informed Variational Autoencoders (CI-VAEs): In addition to the VAE network, a linear discriminator network with trainable weights predicts classes of observations on the latent space, and its classification loss is added to the total loss function of the network model.

classes are linearly separable. The linear discriminator is simultaneously trained with the entire model. The classification loss of the linear discriminator is included in the total cost function of the model. Using MNIST handwritten digit dataset LeCun et al. (1998), we investigated class-specific linear traversal and class-specific synthetic data generation and illustrated how it provided a robust solution as compared to VAEs for data generation during linear latent space traversal. As another use case, we studied Colon Cancer single-cell RNAseq data Qi et al. (2022) using CI-VAE to enhance our understanding of the development of cancer cells from normal colon cells (See Appendix).

## 2 AUTOENCODERS

Autoencoders are unsupervised neural network models, composed of an encoder that provides a deterministic mapping of data $x$ into a lower-dimensional latent space $z$, followed by a decoder to reconstruct the data into its original form $\hat{x}$ (see figure 2a). Among the applications of autoencoders are dimensionality reduction, data compression as well as data denoising Simonyan & Zisserman (2014); Kramer (1991). The total cost function in Autoencoders are the reconstruction error, described by the distance between original input data $x$ and the reconstructed observation $\hat{x}$.

$$J_{AE} = \mathcal{L}(x, \hat{x}) \tag{1}$$

In Variational Autoencoders (VAEs), however, instead of mapping data $x$ to latent variable $z$, we map the input data $x$ to the posterior distribution $p(z|x)$. More specifically, with the assumption of the Gaussian distribution for $p(z|x)$, in VAEs, there is a deterministic mapping of the input data $x$ to the mean $\mu_{z|x}$ and standard deviation $\Sigma_{z|x}$ to construct posterior distribution $p(z|x)$. Followed by random sampling from the posterior $p(z|x)$, latent variable $z$ is constructed. Finally, through the decoder, the latent variable $z$ is deterministically mapped to the reconstructed input data $x$ An & Cho (2015); Kingma & Welling (2019).

The total cost function of the variational autoencoder is comprised of 1) The reconstruction error and 2) The regularization term, enforcing the latent space distribution $p(z|x)$ to be as close as to the prior $p(z) = N(\vec{0}, \vec{I})$ to enact orthogonality/independence of the $z$ axis as well as to provide regularization for the network weights.

This architecture provides the advantage of 1) projecting data into a lower-dimensional space and 2) approximating posterior distribution $p(z|x)$ and 3) enabling the generation of new synthetic data

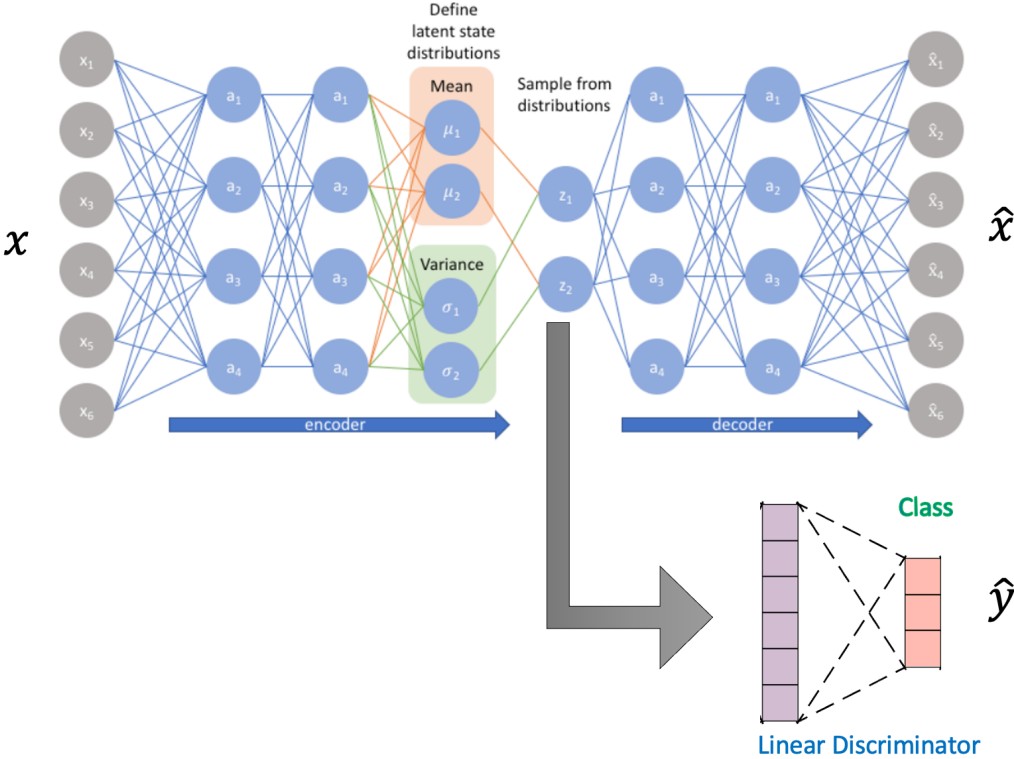

Figure 2: General architecture of Class-Informed Variational Autoencoders (CI-VAEs). The linear discriminator is a single linear layer neural network followed by a softmax function to predict class probabilities of observations on the latent space.

from the inferred latent space.

$$J_{VAE} = \mathcal{L}(x, \hat{x}) + \alpha \times KL[N(\mu_{z|x}, \sigma_{z|x}), N(\vec{0}, \vec{I})] \tag{2}$$

## 2.1 CLASS-INFORMED VARIATIONAL AUTOENCODERS (CI-VAE)

The proposed CI-VAE model is a supervised model based on the VAE architecture with additional components in both the architecture and the total cost function (see figure 2c). To construct CI-VAE, we implemented the current standard VAE architecture with the addition of a linear discriminator as a single layer neural network which is used to map latent variable $z$ to a set of probabilities to predict its class $y$. The linear discriminator is trained with the network with its trainable weights $W_{h \times d}$ where $h$ is the latent space dimension and $d$ is the number of unique classes $y$ of the data $x$. The classification loss (cross-entropy loss) of the linear discriminator is also added to the total loss of the standard VAEs. The linear discriminator loss is thus enforcing observations from different classes on the latent space $z$ to be linearly separable. The aforementioned "Linear" separation is stemmed from the fact that the discriminator network is linear. Thus, the linear discriminator is providing two simultaneous impacts: 1) During the backpropagation, the linear discriminator becomes a better classifier. However, since its performance is plateaued because of its linearity constraint, 2) the gradient signals are sent to the entire model to infer a latent space that is linearly separable based on classes. The entire cost function of CI-VAE is therefore described as:

$$J_{CI-VAE} = \mathcal{L}(x, \hat{x}) + \alpha \times KL[N(\mu_{z|x}, \sigma_{z|x}), N(0, 1)] + \\ \beta \times J_{LD}(y, \hat{y}) \tag{3}$$

where $J_{LD}$ is the cross-entropy loss between the true class $y$ and the linear class $\hat{y}$ predicted by the linear discriminator for any observation $x$.

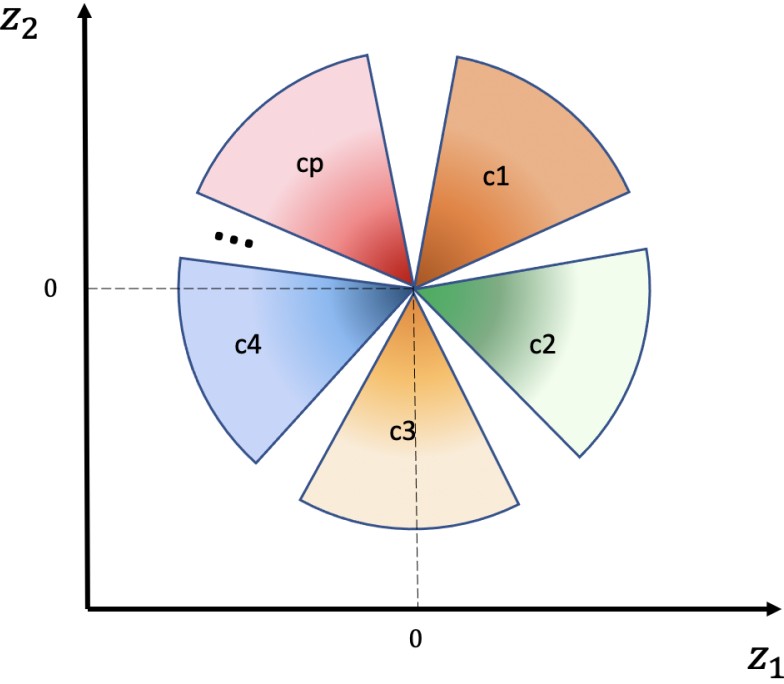

Figure 3: Schematic 2D representation of the latent space $z$ after trained on CI-VAE. Latent variables are normally distributed around 0 with the standard deviation of $I$ enforced by prior $p(z) = N(0, I)$. In addition, enforced by the linear discriminator in CI-VAE, classes are linearly separated.

## 2.2 CLASS-SPECIFIC LINEAR TRAVERSAL AND CLASS-SPECIFIC DATA AUGMENTATION

In figure 3, we demonstrated a schematic representation of the latent space generated by CI-VAE. As illustrated in this figure, there is a higher probability of data around the origin $\vec{0}$. This is imposed by the KL-divergence loss between the posterior $p(z|x)$ and the prior $p(z) = N(\vec{0}, \vec{I})$. In addition, observations from different classes are linearly separable as a result of the linear discriminator network in the CI-VAE model. Linear separability is the desired characteristic when within-class data generation through a linear traversal in the latent space is intended. The linear separability of latent space, theoretically guarantees that any within-class linear traversal would not interfere with the space of other classes. While there is often the class overlapping problem in VAEs, this geometrical property, which is imposed to the latent space in the CI-VAE model, minimizes the chance of interfering with other classes during the within-class data generation through linear traversal.

Linear traversal can be a building block for data augmentation in VAEs. Since CI-VAEs allow for class-specific linear traversal, it enables class-specific data augmentation or in other words, it enables generating synthetic labeled data. To this end, we repeatedly randomly select pairs of observations on the latent space from the desired class and perform linear traversal while generating new samples from the same class.

## 3 EXPERIMENTS

We empirically assessed our proposed CI-VAE on MNIST LeCun et al. (1998) handwritten digits dataset. In addition to training a CI-VAE model, we trained a standard VAE model as a benchmark. For a fair comparison, we used identical model architecture (except for the linear discriminator in CI-VAE), the same hyper-parameters as well as using the same training, validation, and test data.

The MNIST dataset consists of 80,000 handwritten digits images each having $28 \times 28 \times 1$ pixels, with the digits between 0 to 9. We used the digits as the true class associated with each image, although choosing any other arbitrary class in CI-VAE is possible. The data processing included normalizing

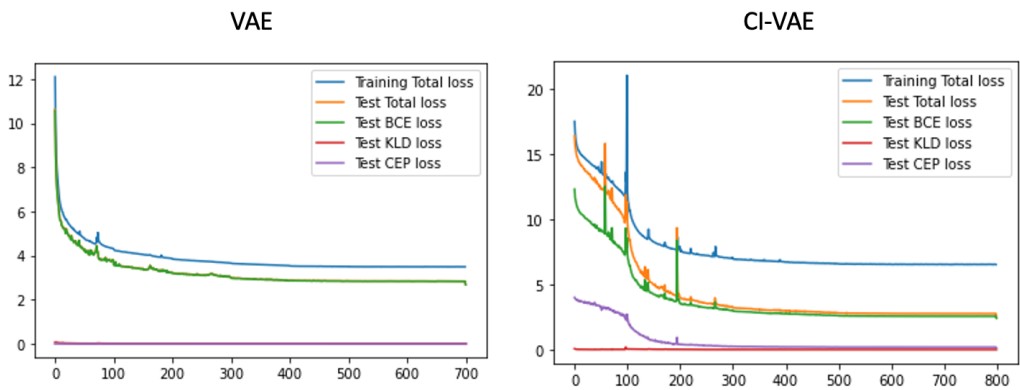

Figure 4: Average loss components v.s. epoch number for **Left**: VAE, and **Right**: CI-VAE

Table 1: Performance comparison between VAE and CI-VAE for reconstruction mean absolute error (MAE) and linear separability of classes on latent space $z$ measured by performance of a linear logistic regression.

|  | VAE | CI-VAE |
|---|---|---|
| Reconstruction Error (MAE) | 2.82 | **2.56** |
| Linear Separability(%) | 90.81 | **99.87** |

pixel values, flattening features into $28 \times 28 = 526$ dimension, and randomly splitting data into 80% for training-set, 10% validation-set, and the remaining 10% for the test-set.

## 3.1 MODEL ARCHITECTURE

The encoder in the CI-VAE model was comprised of seven fully connected layers, each layer is followed by a batch-normalization, relu non-linear activation, and dropout to map input data into the mean and standard deviation vectors of the latent variable $z$ with 20 dimensions. Random sampling is then taken from a Gaussian distribution described by the mean and standard deviation vectors to extract the latent vector $z$. Additionally, the linear discriminator, proposed in CI-VAE, consists of a single fully connected neural network that takes the $z$ vector as input and predicts class probabilities $\hat{y}$ as output. The decoder also consists of seven fully connected layers, with the same layout as the encoder, making deterministic mapping from $z$ to reconstructed $x$ noted as $\hat{x}$. The weights in the model are trained for over 2000 epochs with the batch-size of 512 images until we reached satisfactory convergence (figure 4).

## 3.2 RESULTS

Once CI-VAE and VAE models are trained, we comparatively investigated the following quantities for each model: 1) Reconstruction Quality, 2) Linear separability of classes of observations in the latent space and 3) Quality of class-specific synthetic data generation through linear traversal.

We measured the reconstruction quality using mean absolute error (MAE) between the input data $x$ and the reconstructed data $\hat{x}$. With the measurement of reconstruction error (see table1.), we found out that CI-VAE (with MAE = 2.56) outperformed VAE (with MAE = 2.82). To visualize the CI-VAE and VAE reconstruction performance, in Figure 6, we illustrated a few samples with real digit images, reconstructed images, and the difference-image defined as the simple subtraction of real image from the reconstructed one. As shown, the difference-images are very similar between predictions from CI-VAE and VAE models. We also examined how the latent space in each model was linearly separable based on the classes. To this end, after VAE and CI-VAE are trained and the weights are locked, we trained a separate linear logistic regression classifier applied on the latent space representation of the test data to predict their respective classes (digits in this case study) for both CI-VAE and VAE. Linear logistic regression represents a set of linear hyper-planes that best

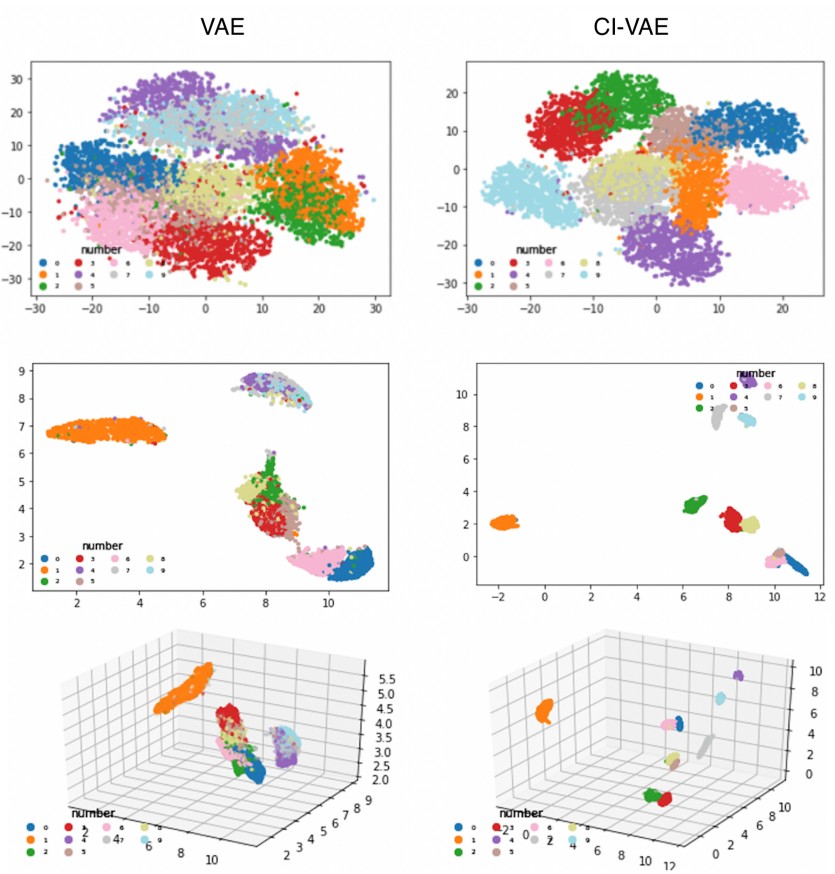

Figure 5: Representation of the latent space on test data for: **Top**: 2D TSNE plot, **Center**: 2D UMAP plot, **Bottom**: 3D UMAP plot for both VAE model on the Left and CI-VAE model on the Right.

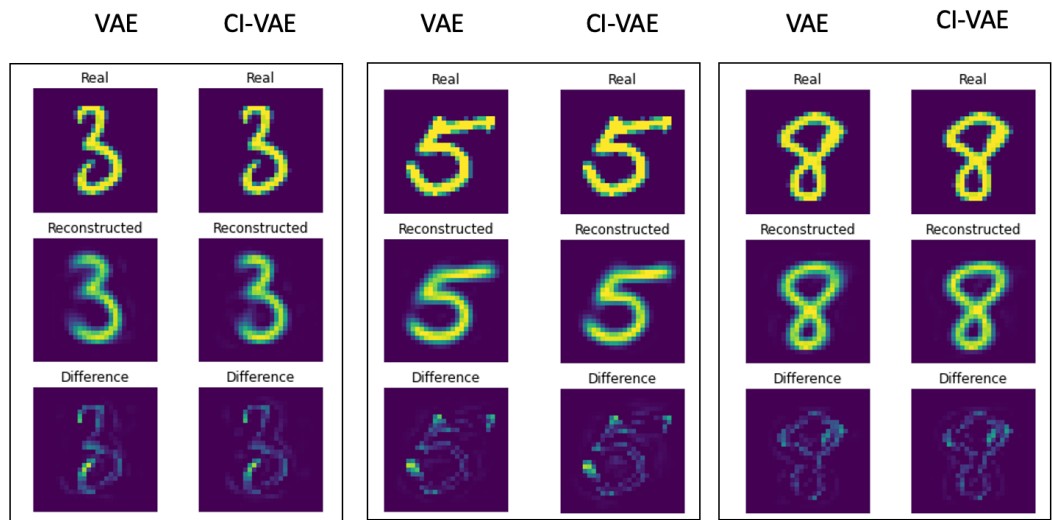

Figure 6: Top: a real observation, Middle: corresponding reconstructed observation, Bottom: error profile. Generated by VAE model (left) and CI-VAE model (right).

discriminate classes on the latent space. With this regression analysis, we showed that CI-VAE achieved much higher classification accuracy (99.87%) compared to VAE (90.81 %) (see table 1), demonstrating that CI-VAE has greatly separated classes in the latent space compared to VAE by nearly 10%. Enforcement of linear separation of classes in the latent space in CI-VAE enhanced the performance of within-class data generation through linear traversal. In figure 7, we illustrated three examples where we aimed to generate within-class data through latent space linear traversal in both CI-VAE and VAE. We performed this task for digits 3, 5, and 8 respectively. As shown in figure 7, in each traversal, VAE generated figures resembling a different digit that was originally intended. For example, VAE generated an image of digit 8 during traversal within two digits 3 and generated images of digit 6 while traversing within two digits 5 and two digits 8. However, CI-VAE is robustly generating samples of digits fully consistent with the intended digit in all the experiments. This is due to the fact that classes in the latent space are linearly separated in CI-VAE while having a significant overlap in VAE. Due to limited space, we only showed a few examples, but we can draw many more similar examples where CI-VAE robustly generates within-class data where VAE fails to do that.

Within-class linear traversal could be leveraged as the building block for within-class data augmentation. We generated synthetic data belonging to class $C$ by randomly selecting two points $a \in C$ and $b \in C$ on the latent space and through linear traversal, we can generate new within-class synthetic data. Iterative random selection of pairs of points, linear traversal, and generating synthetic data allows for the generation of an unlimited number of within-class synthetic data. In applications where labeled data is limited, using this method in the CI-VAE model, we are able to generate an extensive amount of new labeled synthetic data. Shown in figure 8, through latent space linear traversal within digit 3, we generated many synthetic samples for the digit 3. The data points illustrated in this figure, are presented using a 2D TSNE plot from the test data in its original coordinates.

## 4 DISCUSSION

We introduced CI-VAE to extend the capabilities of VAEs such that the representation of observations in the lower dimensional latent space becomes linearly separable among classes. We hypothesized that by enforcing linear separability of classes using a linear discriminator, the class overlapping issues would be addressed and thus it possibly allow "class-specific" linear traversal and data interpolation.

Shown in figure 2, we purposefully used a discriminator with no non-linearity so as to infer a linearly separable latent space. One may suggest using a non-linear discriminator instead, but linear discriminator is particularly chosen as I) Linear discriminators on the latent space are equivalent to non-linear high dimensional hyper-planes on the original dimension of the data $x$. II) Once the linear discriminator is saturated and is unable to linearly separate the data in the latent space, through gradient back-propagation, it sends signals to the entire network to contribute to forming a latent space that is more linearly separable while maintaining low reconstruction quality. III) For class-specific data generation, we typically perform linear traversal, thus linear separability of classes allows high-quality within-class data generation.

By visualizing the latent space inferred by VAEs and CI-VAEs as in figure 5, we have shown clear separation of classes in CI-VAEs while there exists a significant overlap among classes for the VAE. It is noted that the ability of CI-VAEs to separate classes would be even more apparent when choosing more sophisticated classes such as assigning class 1 to female hand-written digits and class 2 as otherwise. The point is that CI-VAEs may be applied to any arbitrary class definitions.

In CI-VAE, by adding a new cost term from the linear discriminator to the total cost function of the model as shown in equation 3, we may expect the reconstruction quality to be impacted. In table1, we showed that CI-VAE resulted in higher reconstruction quality compared to VAE. This may be a result of providing class information that may have improved learning of the overall model. However, it may be possible to have lower reconstruction error in VAE due to adding linear discriminator loss in CI-VAE. This may occur especially when class definitions are far from the lower-dimensional structure of the data. However, by tuning $\beta$ in equation 3, we can control the contribution of the linear discriminator loss to maintain high reconstruction quality in CI-VAE models.

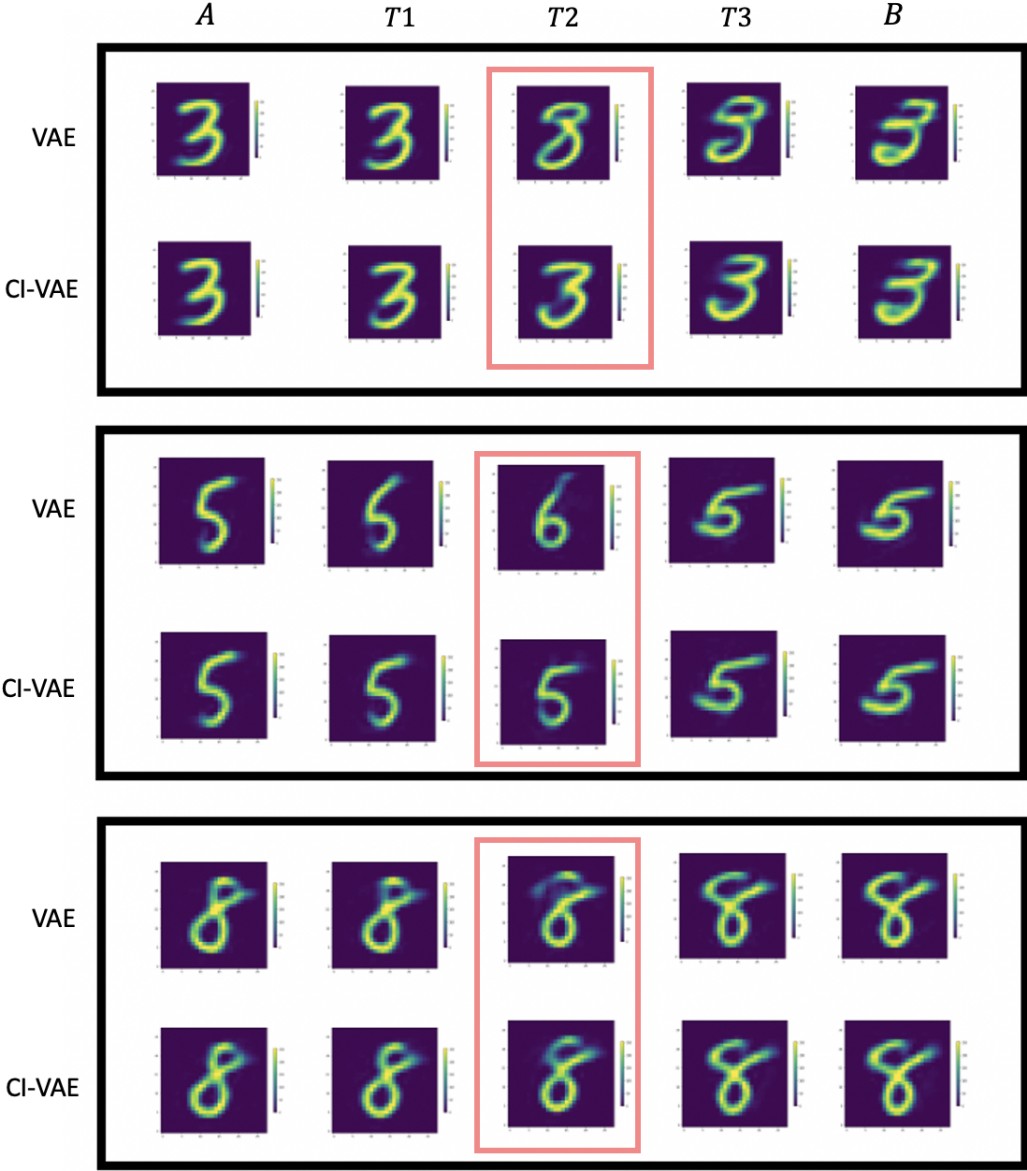

Figure 7: Linear Traversal for three experiments using VAE (top) and CI-VAE (bottom) models. In each experiment, the first and last frames ($A$ and $B$) are real data are also the same across VAE and CI-VAE models. The three middle frames ($T1$, $T2$, and $T3$) are synthetically generated frames during linear traversal for each model. It is shown that for VAE, linear traversal occasionally leads to generating data that may belong to a different class. For example digit 3 is turned into 8 and 9, digit 5 is turned into 6, and digit 8 is turned into 6. However, synthetic data generated by CI-VAE are all robustly within the intended class/digit.

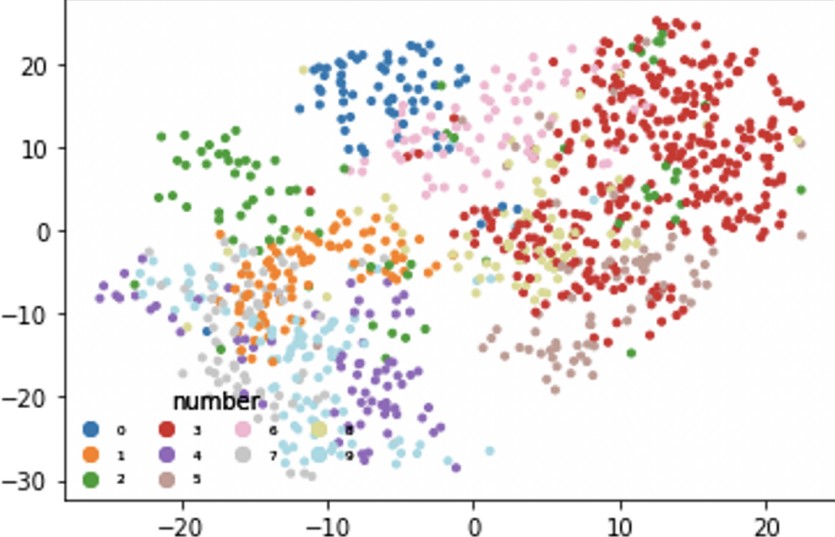

Figure 8: Data augmentation of digit 3 using CI-VAE model shown in 2D TSNE plot on the original dimensions $x$.

By fitting a logistic regression on the latent space, we have shown in Table 1 that the linear separability in CI-VAE has increased by 10%. Thus, as illustrated in figure 7, through linear traversal, VAE is generating images from digits other than intended while CI-VAE robustly generated digits from the intended digit class.

Having linearly separated classes in the latent space, allows to generate class-specific data. In Figure 8, data augmentation of digit 3 is performed using CI-VAE. In VAE however, we are unable to generate samples from a specific class as classes are overlapping on the latent space. Thus, in scenarios where we have limited labeled data, we can use CI-VAE to generate new labeled samples from any class and thus to increase the amount of data to potentially enhance training performance.

CI-VAE may also be used as a semi-supervised learning model Zhu (2005); Chapelle et al. (2009). Labeled portion of the training data can participate in training the entire network, while the unlabeled potion of data may be used to train the model (encoder and decoder) without the participation of the linear discriminator network.

While in CI-VAE, we addressed the issue of overlapping classes in latent space to allow class-specific data interpolation, both VAE and CI-VAE may not address the issue of having possible discontinuity (holes) on the latent space. Though, there are many ongoing research works on addressing this issue.

## 5 CONCLUSION

In this paper, we proposed Class-Informed Variational Autoencoder (CI-VAE), a generative neural network model comprised of an encoder and decoder similar to the standard architecture of variational autoencoder, with an addition of a linear discriminator network which predicts classes of data from its latent representation. This architecture results in the construction of a probabilistic latent space that is also linearly separable among different classes. This enables within-class data generation through latent space linear traversal between two arbitrary samples of the same class. We showed that the addition of a linear discriminator loss into the loss function minimizes the possibility of generating samples from a different class during linear traversal. We have conducted a comparative study between CI-VAE and VAE using the MNIST dataset for handwritten digits. We showed that CI-VAE provides more robust within-class data generation through linear traversal and improved class-specific data augmentation to compare with VAEs. A wide variety of applications requiring within-class traversal for generating synthetic labeled data in various supervised or semi-supervised problems are made possible using CI-VAEs.

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
