# OpenReview forum: "CI-VAE: a Class-Informed Deep Variational Autoencoder for Enhanced Class-Specific Data Interpolation"
_ICLR.cc/2023/Conference — Submitted to ICLR 2023_

### Official Review · Reviewer_ftSd · 2022-10-12

**Confidence:** 5
**Correctness:** 3
**Technical Novelty And Significance:** 1
**Empirical Novelty And Significance:** 1
**Recommendation:** 1

**Clarity, Quality, Novelty And Reproducibility:**

- The paper is well written and very easy to follow.
- Many aspects of the idea have not been as deeply investigates as potentially possible. For example, the experiments are limited to the very common MNIST data set, which is inadequate to fully investigate the utility and performance of the method.
- Also the manuscript does not contain any experiments on semi-supervision.
- The literature review is lacking much if not all relevant previous works. For example, the original semi-supervised VAE [1] isn't even mentioned. In addition, a more recent paper [2] discusses the same idea in great detail.

- There are some weaknesses in the factual correctness of the writing, e.g. the paper by Higgins et. al. is cited as semi-supervised, which seems inadequate. Similarly, the paper states that 28*28 = 526.


[1] - Kingma, D. P., Mohamed, S., Jimenez Rezende, D., & Welling, M. (2014). Semi-supervised learning with deep generative models. Advances in neural information processing systems, 27.
[2] - Joy, T., Schmon, S., Torr, P., Siddharth, N., & Rainforth, T. (2021). Capturing Label Characteristics in VAEs. In International Conference on Learning Representations.

**Strength And Weaknesses:**

_Strengths_:

- The idea is interesting, well motivated and well presented.
- It makes sense to use label information when it is available. Guiding the latent variable using a weak classifier is a sensible regularization strategy.
- Using a weak classifier helps impose structure on the latent space.

_Weaknesses_:

- The paper is lacking experiments in several directions. There is only one experiment in the paper and that experiment is with the common MNIST dataset. The approach doesn't compare to any of the previous semi-supervised approaches.
- While I appreciate the utility of a weak classifier, I am not convinced that linear separability is completely necessary or even desirable. I think a more detailed argument would be necessary.

**Summary Of The Paper:**

The authors introduce a variational autoencoder with associated loss function that enables to supervision of the latent embedding by class information. In particular, the authors consider linear separability preventing overlapping representations for different classes.

**Summary Of The Review:**

- Overall the ideas presented in the paper are useful. The presentation is good, but relatively weak in terms of theory and experiments. The same ideas have been discussed in the literature before and in more detail.

---

### Official Review · Reviewer_W4UK · 2022-10-23

**Confidence:** 5
**Correctness:** 3
**Technical Novelty And Significance:** 2
**Empirical Novelty And Significance:** 2
**Recommendation:** 1

**Clarity, Quality, Novelty And Reproducibility:**

Code availability is not clearly stated.


**Strength And Weaknesses:**

Strength

- The strategy seems straightforward.

Weaknesses

- In the main text, the model was tested only using MNIST against vanila VAE, and lacking quantitative comparison.
- In the appenix, the authors applied teh CI-VAE against a particular colon cancer single cell RNA-seq dataset, but the reason why this particular dataset was chosen is unclear, and there is a huge jump from MINST.


**Summary Of The Paper:**

The authors proposed Class-Informed Variational Autoencoder (CI-VAE), a generative neural network model comprised of an encoder and decoder similar to the standard architecture of variational autoencoder, with an addition of a linear discriminator network which predicts classes of data from its latent representation. The model was tested with MNIST against vanila VAE.


**Summary Of The Review:**

The authors proposed Class-Informed Variational Autoencoder (CI-VAE), a generative neural network model comprised of an encoder and decoder similar to the standard architecture of variational autoencoder, with an addition of a linear discriminator network which predicts classes of data from its latent representation. The proposed model was examined only using MNIST and comparison is mostly qualitative.

---

### Official Review · Reviewer_29ie · 2022-10-24

**Confidence:** 3
**Correctness:** 4
**Technical Novelty And Significance:** 3
**Empirical Novelty And Significance:** 2
**Recommendation:** 6

**Clarity, Quality, Novelty And Reproducibility:**

Very clear and reproductible.


**Strength And Weaknesses:**

The paper is very written, clear, and sound. The main weakness is that there is a lack of comparison with other ML approaches for interpolation and a discussion about how the approach differs from the disentangled VAE approaches.

In particular, one important approach to performing class-specific interpolation can be found in the Optimal Transport ML literature. How does OT interpolation compare to CI-VAE interpolation for MNIST? What would be the benefit/drawback of such an approach versus OT? (The versatility of the CI-VAE is one benefit, but a thorough discussion would be very welcome).

For biology, there is recent literature about interpolation with OT that the authors should cite and look into. For instance (but not limited to) see and references therein:
- Fast and Smooth Interpolation on Wasserstein Space
- TrajectoryNet: A Dynamic Optimal Transport Network for Modeling Cellular Dynamics


**Summary Of The Paper:**

This paper proposes adding a linear discriminator in the training loss of VAE in order to ensure separability of the classes. The goal of such an approach seems to either increase the VAE performance or allow linear interpolation between the data class, which hence provide a quite versatile approach to perform interpolation between object of the same class.



**Summary Of The Review:**

The main drawback of the paper is the lack of discussion and comparison with concurrent approaches, which make the importance of the paper not clear enough.

---

### Official Review · Reviewer_RrTY · 2022-10-26

**Confidence:** 4
**Correctness:** 3
**Technical Novelty And Significance:** 1
**Empirical Novelty And Significance:** 2
**Recommendation:** 1

**Clarity, Quality, Novelty And Reproducibility:**

The paper is easy to follow, however, some details are missing to ensure full reproducibility of the paper.

**Strength And Weaknesses:**

Strengths:
- The paper is clearly written and easy to follow.
- The proposed method seems to achieve the goal of linear class separability in the latent code.

Weaknesses:
- The paper is missing important baselines and comparisons to e.g. class-conditional general or GMM-VAEs. Furthermore, it is lacking any discussion of previous work on latent space shaping such as [1]. [1] seems very related and also enforces class separation and is only a single example for missing literature overview.
- It is not clear whether, linear interpolation is the best or correct interpolation method, instead spherical interpolation could be used. [2] has an overview of different interpolation methods. This choice should be better justified.
- The paper seems unfinished and unpolished in places - it mentions results on cancer genomics data in the abstract but it seems to only show results on MNIST, which as a dataset is not enough to warrant publication.

Misc:
- Please correct the usage of `\citep` and `\citet` for the ease of reading.
- Most Figure references seem to be incorrect.
- Around eq. 1 it is mentioned that AEs optimise the distance between original data and reconstructed data. It would be good to be specific and add example distance metrics.
- The introduction of VAEs is a bit imprecise when it comes to the likelihood of the output space.
- Fig. 3 shows a schematic example latent space for the CI-VAE. However, the logistic regression is applied to the samples of z ~ p(z|x) which means that there is no guarantee that z is not sampled outside of the class boundary.
- Sec 2.2 describes some usage of the model for data augmentation. However, it is very imprecise and should compare to class-conditional VAEs.
- Fig. 4 shows the losses but it is unclear which loss is which.
- p 5: 28*28 = 784 instead of 28*28=526
- Table 1: It would be good to add error bars.
- Why did you use dropout in this model?
- Figure 5: How do you interpret these latent space plots?
- When you perform linear separability testing, how is the new classifier trained? What is the reported value tested on?
- I do not understand the value of figure 8.
- What values are chosen for $\alpha, \beta$?

[1] Connor, Marissa, Gregory Canal, and Christopher Rozell. "Variational autoencoder with learned latent structure." International Conference on Artificial Intelligence and Statistics. PMLR, 2021.
[2] Mi, Lu, et al. "Revisiting Latent-Space Interpolation via a Quantitative Evaluation Framework." arXiv preprint arXiv:2110.06421 (2021).

**Summary Of The Paper:**

The paper proposes to apply a logistic regression to the latent space samples of a VAE to enforce linear separability in the latent code. The authors argue that this approach is useful for interpolating between two samples of the sample class without crossing class boundaries. The experiments on MNIST indicate that the regularised latent space fulfils the set-out requirements better than a vanilla VAE.

**Summary Of The Review:**

The paper seems generally unpolished with references to missing experiments and incorrect figure references. A lot of relevant background literature is missing which also leads to insufficient experiments and baselines for them.

---

### Decision · Program_Chairs · 2023-01-20

**Decision:**

Reject

**Justification For Why Not Higher Score:**

The paper is a clear reject.

**Justification For Why Not Lower Score:**

N/A

**Metareview: Summary, Strengths And Weaknesses:**

Although the reviewers appreciated aspects of the paper (clarity and soundness), the overall consensus is that the proposal of introducing class information in the latent is not sufficiently novel to merit publication of the paper.